# First Report of a Case of Ocular Infection Caused by *Purpureocillium lilacinum* in Poland

**DOI:** 10.3390/pathogens10081046

**Published:** 2021-08-18

**Authors:** Robert Kuthan, Anna K. Kurowska, Justyna Izdebska, Jacek P. Szaflik, Anna Lutyńska, Ewa Swoboda-Kopeć

**Affiliations:** 1Department of Medical Microbiology, Medical University of Warsaw, 5 Chałubińskiego Street, 02-004 Warsaw, Poland; 2Department of Ophthalmology, Medical University of Warsaw, 24/26 Marszałkowska Street, 00-576 Warsaw, Poland; anna.kurowska@wum.edu.pl (A.K.K.); justyna.izdebska@wum.edu.pl (J.I.); jacek.szaflik@wum.edu.pl (J.P.S.); 3Public Ophthalmic Teaching Hospital, 24/26 Marszałkowska Street, 00-576 Warsaw, Poland; 4Department of Medical Biology, National Institute of Cardiology Stefan Cardinal Wyszyński State Research Institute, 42 Alpejska Street, 04-628 Warsaw, Poland; anna.lutynska@ikard.pl (A.L.); e.swoboda@ikard.pl (E.S.-K.)

**Keywords:** *Purpureocillium lilacinum*, *Paecilomyces lilacinus*, keratitis, voriconazole, Poland

## Abstract

This report describes the first case of an ocular infection induced by *Purpureocillium lilacinum* in Poland. The patient was a 51-year-old immunocompetent contact lens user who suffered from subacute keratitis and progressive granulomatous uveitis. He underwent penetrating keratoplasty for corneal perforation, followed by cataract surgery due to rapid uveitic cataract. A few weeks later, intraocular lens removal and pars plana vitrectomy were necessary due to endophthalmitis. The patient was treated with topical, systemic, and intravitreal voriconazole with improvement; however, the visual outcome was poor. The pathogen was identified by MALDI-TOF MS.

## 1. Introduction

The reported prevalence of ocular fungal infections has increased substantially over the past decades, as microbiology diagnostic capabilities have improved [1,2]. Fungi can cause both endogenous and exogenous keratitis, keratomycosis, and/or endophthalmitis. Classical diagnosis of fungal infections mainly comprises a direct microscopic examination of clinical material, isolation of the fungus in pure culture, taxonomic identification, and determination of in vitro antifungal agent susceptibility [3,4,5]. Thus, the identification of fungi using conventional techniques generally requires 3–7 days [6]. To improve identification, the use of molecular biology methods, such as real-time polymerase chain reaction (RT-PCR), for pathogen DNA quantification in specimens, can differentiate between true infection and possible contamination of the anterior chamber by microorganisms present in the conjunctival flora [7,8,9,10]. Moreover, the results of the PCR identification are available in less than 24 h [11]. Other diagnostic methods, including in vivo confocal microscopy, can elucidate the ocular pathology at the cellular level [12,13,14].

Matrix-assisted laser desorption/ionization time-of-flight mass spectrometry (MALDI-TOF MS) is a relatively new tool in medical microbiology. In many clinical laboratories, MALDI-TOF MS has become the standard method for routine identification of many pathogens, replacing the various biochemical assays that were previously routinely needed. However, in the case of mycology, especially for filamentous fungi, MALDI-TOF MS has not yet been fully adapted, verified, and/or validated, mainly due to the complex protocols required for handling samples [15].

Ocular fungal infections have been reported to be caused by more than 56 genera. Among these, the most commonly involved are *Candida* and *Aspergillus*; other unicellular and filamentous fungi such as *Fusarium*, *Penicillium, Paecilomyces*, *Acremonium*, *Exophiala*, *Pseudallescheria*, *Scytalidium*, *Sporothrix*, and *Cryptococcus*; and dimorphic fungi [12,16,17,18,19,20].

Most filamentous fungi are saprophytic microorganisms that are found in a variety of natural habitats. Human exposure to fungus-contaminated plants or other organic matter may facilitate exogenous infections of the eye, resulting in corneal trauma [21,22,23].

The use of hard or soft contact lenses is recognized as a predisposing factor for the development of keratitis of bacterial (mainly *Pseudomonas aeruginosa*) and/or simultaneous bacterial and fungal (mainly *Candida* species) origin [24].

*Candida*, *Fusarium*, and *Aspergillus* species are the most frequently isolated etiological agents of ocular fungal infections, commonly resulting in the development of endophthalmitis or keratitis [5,25,26,27,28]. *Paecilomyces* species have also been described as the etiological agents of various infections in humans, including eye infections [29]. The hyphomycete genus of *Paecilomyces* was introduced into fungal taxonomy by Bainier in 1907. Phylogenetic studies of the genus resulted in major reclassification, including the transfer of *Paecilomyces lilacinus* to a new genus as *Purpureocillium lilacinum* [30]. Further detailed morphological studies coupled with the sequencing of four different taxonomically important loci of clinical and environmental isolates identified *Purpureocillium lavendulum* as a new species within the genus *Purpureocillium.*

*Paecilomyces* species are common environmental molds. They can be isolated from soil, decomposing plants, fruit, vegetables, insects, and nematodes [31]. *P. lilacinum* is a nematophagous fungus that can target eggs, juveniles, and females, thus reducing the soil plant-parasitic nematode population. This has led to *P*. *lilacinum* being described as a ‘green’ pesticide that is useful in commercial and amateur horticulture.

*P. lilacinum* and other related species have seldom been associated with human infections [32]. However, *Paecilomyces variotii*, *Paecilomyces marquandii*, and *P. lilacinum* are regarded as emerging causes of mycotic keratitis and hyalohyphomycosis, especially in immunocompromised patients. Over 70% of cases of keratitis caused by *Purpureocillium* species have been accompanied by soft contact lens use. Other infections potentially caused by *Purpureocillium* species include those of the nasal sinus, skin, and soft tissue [33,34,35,36].

A new species, *Purpurecillium roseum,* identified by morphological and molecular phylogenetic studies, has recently been isolated from the human cornea. *P. roseum* was found to have similar properties to *P. lilaciunum,* including resistance to amphotericin B and itraconazole, and susceptibility to voriconazole [37].

From a medical perspective, it is notable that, unlike *P. variotii*, *P. lilacinum* is not susceptible to amphotericin B and ‘first-generation’ triazoles (e.g., fluconazole or itraconazole); however, it is susceptible to ‘second-generation’ triazoles such as voriconazole, posaconazole, and ravuconazole [31,38].

This study reports a case of *P. lilacinum*-induced keratitis in a contact lens user.

## 2. Materials and Methods

### 2.1. Culture

Clinical specimen fluid from the anterior chamber of the eyeball and the inflammatory membrane of the surface of the lens were cultured on classical media used for fungi and bacteria (e.g., liquid Sabouraud, brain heart infusion broth, and Schaedler broth, respectively). After 10 days of incubation at 37 °C and subsequent subculture on solid media (Columbia agar with 5% sheep blood, Schaedler agar with vitamin K1 and 5% sheep blood, and chocolate agar), no bacterial growth was detected. All media were purchased from Becton Dickinson, Heidelberg, Germany.

After five days of culture, growth was noticed in the Sabouraud liquid broth. Subsequently, the broth was divided into two portions in Falcon™ 15 mL conical centrifuge tubes (Thermo Fisher Scientific, Waltham, MA, USA) and centrifuged at 5000 rpm for 5 min. The sediment from a single tube was transferred onto a solid Sabouraud medium and incubated for several days at 30 °C until aerial mycelium had been formed; these were subjected to identification based on the detailed analysis of colony pigmentation, morphology of the hyphae, and the shape/size of conidiophores. Microscopic examination was performed with the use of an Olympus CH20BMF200 microscope. The analysis was performed according to the recommendations of the Mycology Online website of the University of Adelaide [7].

Simultaneously, a second portion of the sediment taken from the second tube was subjected to MALDI-TOF MS. The analysis was performed using the VITEK^®^ MS system version 3.0 (bioMérieux; Marcy, l’Etoile, France). Briefly, 1 µL of the inoculation loop mycelium was transferred onto a disposable target plate, air-dried, and treated with 2 µL of 70% formic acid solution followed by the application of 1 µL of the matrix (α-cyano-4-hydroxycinnamic acid dissolved in 50%/50% (*v*/*v*) aqueous acetonitrile containing 2.5% (*v*/*v*) trifluoroacetic acid (bioMérieux; Marcy, l’Etoile, France), then air-dried, and finally analyzed. Mass spectra within a range of 2000 to 20,000 Da were recorded in linear positive mode at a frequency of 50 Hz by a 337 nm nitrogen laser with a fixed focus. For each interrogation, laser shots at different positions within the target well produced up to 100 mass profiles. As reliable results were not obtained, a new sample was prepared with the following modifications: instead of formic acid, the sample was treated with 2 µL of 70% trifluoroacetic acetic acid (bioMérieux; Marcy, l’Etoile, France), air dried, overloaded with 1 µL of the matrix, and analyzed. Testing was performed in duplicate.

### 2.2. Drug Susceptibility

The drug susceptibility tests were performed on Roswell Park Memorial Institute (RPMI) medium (bioMérieux, Marcy, l’Etoile, France). The MIC values were determined for voriconazole, itraconazole, and amphotericin B using the E-test^®^ method. For interpretation of the MICs, the *CLSI Reference Method for Broth Dilution Antifungal Susceptibility Testing of Filamentous Fungi* (3rd edition) was used, supported by the findings of Borman et al. [39,40].

### 2.3. Results

The VITEK MS analysis confirmed the presence of *P. lilacinum* (confidence value = 99). The mass spectra characteristic for *P. lilacinum* are presented in Figure 1. The fungal pathogen responsible for infection was susceptible to voriconazole (0.032 mg/L), resistant to itraconazole (32.0 mg/L), and resistant to amphotericin B (32.0 mg/L).

## 3. Results

A 51-year-old male patient, a lawyer by profession, was referred to the Department of Ophthalmology, Medical University of Warsaw because of unsuccessful treatment of keratitis in his left eye. About 4 weeks earlier he presented with pain, redness, and blurred vision. Due to myopia and astigmatism, he regularly used daily wear soft contact lenses. He denied eye injury, but he noted a short episode of acute pain and foreign body sensation during visitation of a new school building a few days before symptoms had begun. Mild ciliary injection, superficial corneal ulcer, and reduced corneal sensation were noted. Generally, the patient was found to be healthy and a non-smoker. In vivo corneal confocal microscopy (IVCCM) at that moment excluded *Acanthamoeba* cysts and filamentous fungus. However, activated keratocytes and inflammatory cells were observed in corneal tissue. According to clinical findings and IVCCM results, herpetic keratitis was diagnosed, and the patient was treated with topical and systemic acyclovir and topical 0.3% ofloxacin and cefuroxime (2 times 500 mg per day for 7 days) as prophylaxis of bacterial coinfection. After a few days, grayish infiltration deep in stromal tissue was observed, and empirical treatment such as 1.0% voriconazole and 0.2% fluconazole eye drops were prescribed (6 times daily each), as well as systemic fluconazole (2 times 200 mg on the 1st day and thereafter 2 times 100 mg per day) was added. Within the next 2 weeks, keratitis intensified, and inflammatory precipitates appeared on the corneal endothelium and in the anterior chamber. Systemic methylprednisolone at the dose of 16 mg per day was added, and the patient was referred to the hospital.

On admission, best-corrected visual acuity (BCVA) on the Snellen chart was 1.0 for the right and 0.1 for the left eye. Intraocular pressure (IOP) was 15 and 28 mmHg for the right and left eye, respectively. On slit lamp examination, mild lid edema, moderate mixed conjunctival injection, and chemosis were observed (Figure 2).

The central cornea was involved, and greyish infiltrate in the posterior stroma with irregular, ‘feathery’ edges was observed (Figure 3).

The corneal epithelium surface was irregular but without ulceration. The paracentral and peripheral cornea were normal. Inflammatory precipitates on the corneal endothelium and hypopyon in the anterior chamber were present. IVCMC revealed linear structures with characteristic dichotomous branching pattern of fungal hyphae in the corneal stroma (Figure 4).

Based on clinical and confocal microscopy findings, a filamentous fungus was strongly suspected as an etiological factor, and filamentous fungal keratitis and moderate anterior uveitis were diagnosed. The patient was treated topically with 1.0% voriconazole every 2 h, 0.3% ofloxacin (5 times per day), and Dicortineff 4 times per day (2500 IU, neomycin, 25 IU gramicidin, and 1 mg fludrocortisone per 1 mL). Oral voriconazole (2 times 400 mg on the first day, followed by a dose of 200 mg 2 times per day) was started in association with dexamethasone i.v. (2 times 6 mg for 10 days) followed by methylprednisolone (24 mg per day p.o.) and aciclovir (5 times 400 mg).

After initial resolution of stromal infiltrate and symptoms of anterior uveitis, central corneal thinning was observed, followed by corneal perforation (Figure 5).

Emergency penetrating keratoplasty (PK) was performed. In the early postoperative period, the BCVA improved to 0.5 on the Snellen chart, and the IOP was well controlled with anti-glaucoma medication. Topical 1.0% voriconazole 6 times daily, 0.3% ofloxacin ointment (4 times per day), and 0.5% loteprednol (4 times per day) were prescribed. Systemic prednisone (20 mg per day) and aciclovir (5 times 400 mg per day) were continued, but as fungal etiology was not confirmed, systemic voriconazole (2 times 200 mg per day) treatment was continued only for 2 weeks after PK.

Three weeks after surgery, the patient presented with moderate ciliary injection and mild ocular pain. BCVA was 0.5 on the Snellen chart, and the corneal graft was clear. However, iris nodules near the pupillary margin were observed at 11.00 and 12.00 o’clock positions, and fibrinous vascularized membrane adherent to the iris and the anterior lens capsule in its upper part (Figure 6).

Within the next 5 days, the clinical signs were getting worse, and the fibrinous membrane covered the whole surface of the anterior capsule. The flaremetry readings were 158.2 ± 8.3 photon counts (PC)/ms. The lens was swollen and nearly totally opaque (Figure 7).

BCVA of his left eye was 0.1 on the Snellen chart, and the eye fundus could not be examined due to cataract. Ultrasound of the eye (USG) was negative for vitreal exudates. The patient was in general good condition and there were no signs of systemic infection; white blood cell count was 7.89 × 10^9^/L, the cells were predominantly neutrophils (76.9%), erythrocyte sedimentation rate (ESR) was 9 mm/h, and CRP was 0.9 mg/L. The other routine laboratory test results were normal. Additional tests for tuberculosis, syphilis, Lyme disease, HIV infection, and ocular sarcoidosis as potential infectious or non-infectious etiology of granulomatous uveitis were negative. Systemic voriconazole was prescribed again at a dose of 200 mg 2 times per day. The patient underwent phacoemulsification with intracapsular lens implantation in his left eye. The culture of aqueous humor and fibrous membrane from the anterior iris surface was positive for *P. lilacinum*, sensitive to voriconazole, and resistant to itraconazole; details are given in Section 2—Materials and Methods. The culture for aerobic, anaerobic, and yeast revealed all negative results. The microscopic appearance of *P. lilacinum* is presented in Figure 8, Figure 9 and Figure 10. The image of cultures of *P. lilacinum* on the Sabouraud agar is presented in Figure 11.

Five weeks after cataract surgery, the patient presented with blurred vision and ocular pain. BCVA was 0.05 in his left eye, with an IOP of 32 mmHg. The corneal graft was transparent. A dens cellular exudate in the anterior chamber (Figure 12), fibrinous membrane on the artificial lens (Figure 13), and a haze in the vitreous body (vitritis) were visible upon slit lamp examination.

Fundus examination was not possible due to an important vitritis. B-scan ultrasonography confirmed pronounced vitreous opacities and excluded retinal or choroidal detachment. The diagnostic pars plana vitrectomy (PPV) associated with voriconazole intravitreal voriconazole (100 µg/0.1 mL saline) injection and anterior chamber irrigation were performed. The intraoperative examination revealed white masses behind the artificial lens inside the lens capsule, which suggested the presence of intraocular fungal elements. *P. lilacinum* was again cultured from the fibrinous membrane and aqueous humor. A few days later the patient underwent intraocular lens and the capsular bag explantation and 25-gauge pars plana vitrectomy and silicone oil tamponade. The antifungal therapy was continued both topically and systemically for the next 3 months (the total systemic voriconazole treatment period was 6 months). At the 6-month follow-up, there was no recurrence of keratitis or uveitis, and the silicone oil was removed. Currently, the patient is being treated with antiglaucoma medication, and corneal retransplantation due to graft opacification and secondary intraocular lens implantation is planned.

## 4. Discussion

*P. lilacinum* infections are challenging to diagnose and treat. In many cases, classical direct microscopy of Gram-stained material taken during vitrectomy or corneal scrapings is insufficient to detect fungal structures or wrongly identifies yeast-like cells [41].

The application of MALDI-TOF MS in microbiological diagnostics significantly shortens the time needed for pathogenic bacteria or fungal identification [42,43,44,45]. Nevertheless, much care is needed with sample preparation to overcome identification failures. Rychert et al. [46] showed a failure of *P. lilacinum* identification using VITEK MS version 3.0 in two out of 31 strains tested in comparison with DNA sequencing, with no clear explanation for the phenomenon. Similarly, in our report, standard sample procedures with the use of acetic acid extraction did not result in proper identification of the isolate, although this was possible after implementing the methodology modifications described.

Systemic and ocular immunosuppression are the main risk factors for ocular fungi infections [47]. The main factors predisposing keratitis development with *P. lilacinum* etiology described by Yuan et al. [48] and Chen et al. [49] involve pre-existing corneal disease or history of previous ocular surgery, corneal trauma, contact lens use, and unknown risk factors. In this case, both the coexistence of corneal micro-damage and contact lens usage should be considered. Other issues regarding clinically unrecognized factors might include amphotericin B-resistance of *P. lilacinum*, similar to that of azoles such as fluconazole or itraconazole. Many authors have suggested that even appropriate fungal drugs might not provide the desired therapeutic benefits in a short period of time, resulting in their early exclusion or treatment changes [50,51,52,53,54]. Such phenomena might also be attributed to previous steroid therapy. According to a study by Ali et al. [55], a lack of response to an antibacterial or anti-amoebic treatment in cases of infective keratitis can indicate voriconazole as the drug of the choice. At present, the protocol for voriconazole therapy has not been fully recognized with regard to both duration and dosage. Todokoro et al. [56] pointed out that in cases of infection limited only to the superficial corneal stroma, local voriconazole should be the treatment of choice.

Early microbiology testing is regarded as a key factor for reliable etiology identification and treatment choice. Data in the literature strongly suggest that therapy success and limitation of infection are directly related to the timing of the implementation of voriconazole treatment. However, its efficiency has not been clearly shown, especially in cases when voriconazole therapy has been initiated relatively soon after the onset of clinical symptoms (2–3 weeks). In such situations, the period of treatment has been notably shorter, as infiltration has been limiting within 2–3 weeks, so the therapy has rarely been used for longer than 3 months [50,57].

In cases when bacteria were initially detected as the primary pathogen or other anti-fungal drugs with no proven activity against *P. lilacinum* were used, treatment has generally been long-lasting (between 3 and 16 months), given locally or per os, and has required surgical intervention with partial/total keratoplasty, partial/total vitrectomy, removal of the intraocular lens, and/or corneal transplantation [12,14,15].

Notably, late implementation of other anti-fungal drugs against *P. lilacinum* without voriconazole has led to the progression of infection and treatment failures with immunosuppressed state, local corticosteroid or antimicrobial therapy, contact lens usage, operational procedures, and chronic keratopathy being among the highest risk factors [33,58].

So far, no cases of resistance to voriconazole have been reported in *P. lilacinum* responsible for eye infection during long-term local or per os therapy. However, cases of voriconazole treatment failure and posaconazole therapy success have been reported in *P. lilacinum* eye infections [53]; the latter might therefore be an alternative for patients who cannot be treated with voriconazole or for whom voriconazole treatment does not have positive effects [52,59]. Resistance to voriconazole in vitro was reported in only a single case among 43 *P. lilacinum* strains tested [39]. The application of MALDI-TOF MS to recognize the pathogens involved enabled their identification 2 days earlier compared to classical methods and confirmed the usefulness of the approach.

## 5. Conclusions

In any keratitis case with suspected fungal etiology, where clinical material does not identify a pathogen under preliminary microscopy or culture, and clinical symptoms are not characteristic of *Aspergillus* or *Fusarium* species infections, voriconazole should be used as the drug of choice, and treatment should be continued to allow any fungal etiological factors to be identified. If a fungal pathogen cannot be cultured and/or identified for up to 14 days, therapy should be maintained until signs of pathologic changes are no longer apparent, as confirmed by ophthalmology testing/slit lamp/confocal microscopy. In all such cases, discontinuation of local immunosuppressive therapy and replacement by alternatives, such as 0.5% cyclosporine droplets, should be considered [51,55].

Although microbiology was key to establishing the etiology and drug sensitivity in this case report, the application of MALDI-TOF MS enabled the precise recognition of the pathogen involved 2 days earlier compared to classical methods.

## Figures and Tables

**Figure 1 pathogens-10-01046-f001:**
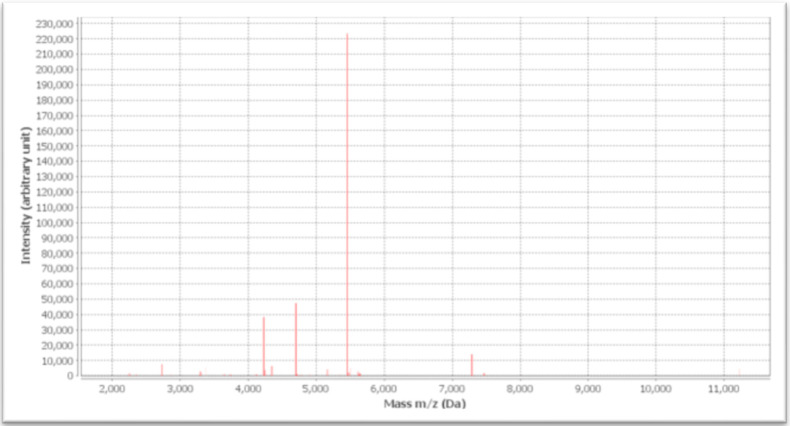
The Vitek MS^®^ MALDI-TOF MS spectrum characteristic for *P. lilacinum*.

**Figure 2 pathogens-10-01046-f002:**
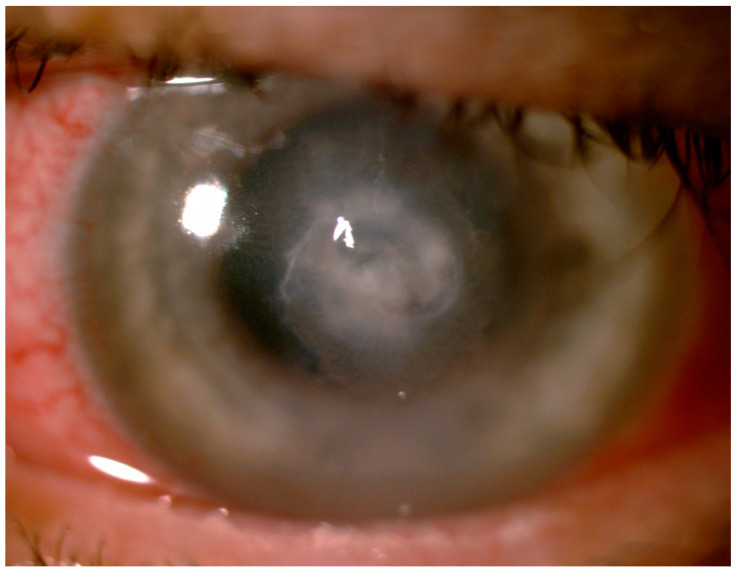
The slit lamp examination of the left eye: mild lid edema, moderate mixed conjunctival injection and chemosis, and central keratitis.

**Figure 3 pathogens-10-01046-f003:**
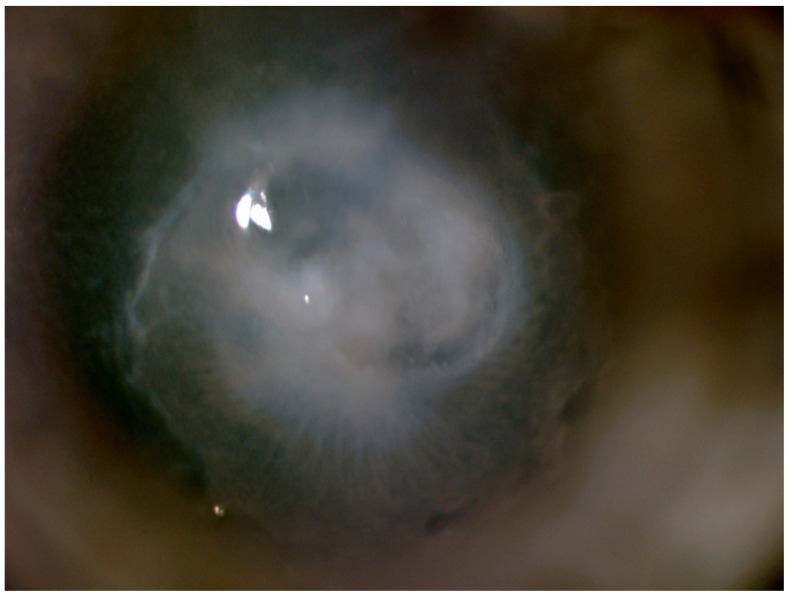
Fungal keratitis: deep stromal infiltrate with feathery edges.

**Figure 4 pathogens-10-01046-f004:**
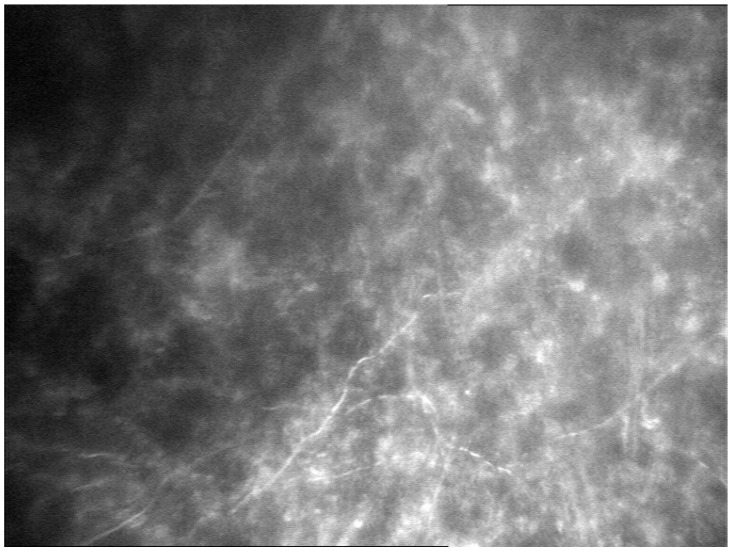
In vivo confocal microscopy: linear structures with a dichotomous branching pattern of fungal hyphae in the corneal stroma.

**Figure 5 pathogens-10-01046-f005:**
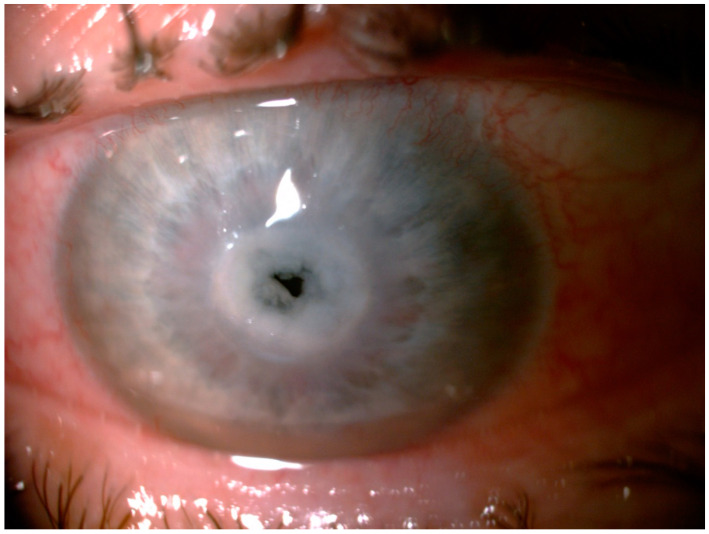
Central corneal perforation.

**Figure 6 pathogens-10-01046-f006:**
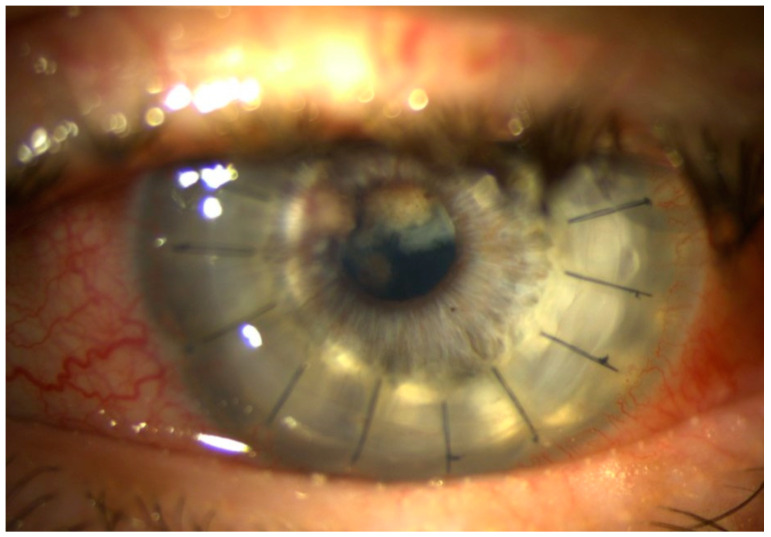
Iris nodules at 11.00 and 12.00 o’clock and fibrinous vascularized membrane adherent to the iris and anterior lens capsule.

**Figure 7 pathogens-10-01046-f007:**
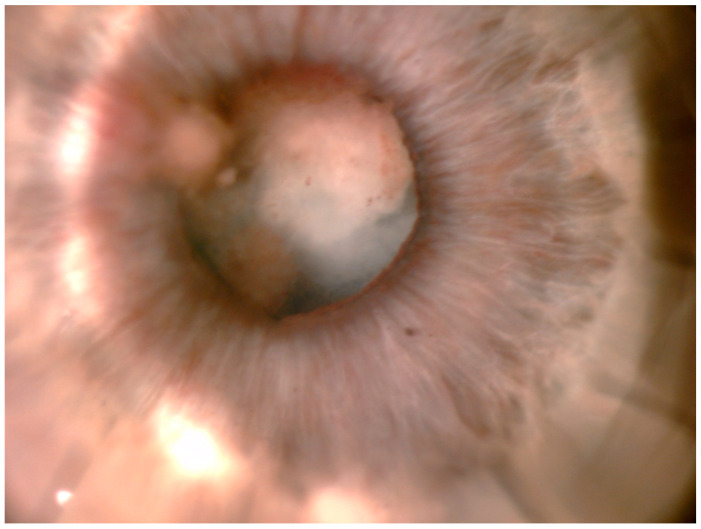
Mature cataract and inflammatory fibrous membrane in the pupil.

**Figure 8 pathogens-10-01046-f008:**
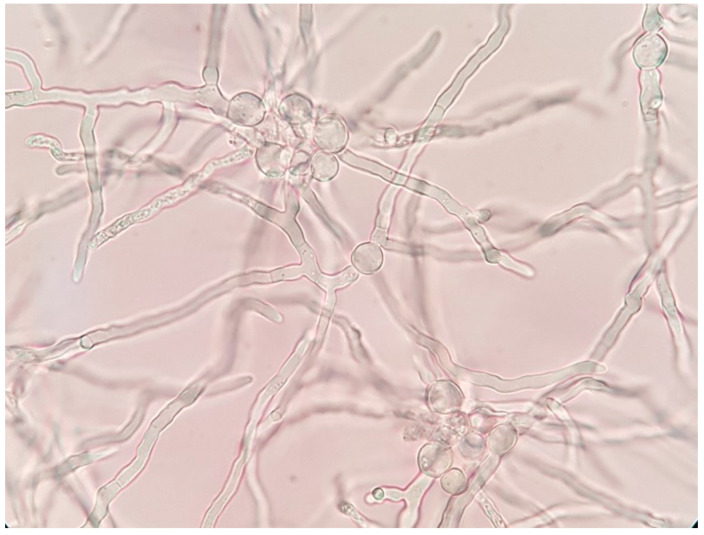
Microscopic appearance of *P. lilacinum* isolated from the liquid Sabouraud culture. Magnification × 1000.

**Figure 9 pathogens-10-01046-f009:**
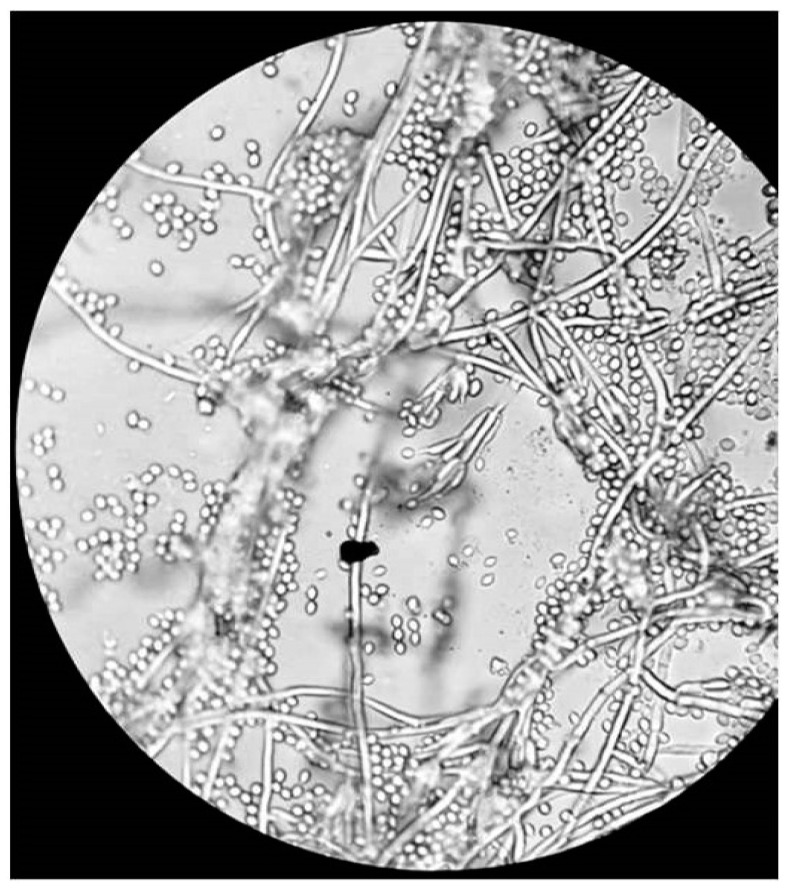
Microscopic appearance of *P. lilacinum* isolated from culture on Sabouraud agar. Magnification × 1000.

**Figure 10 pathogens-10-01046-f010:**
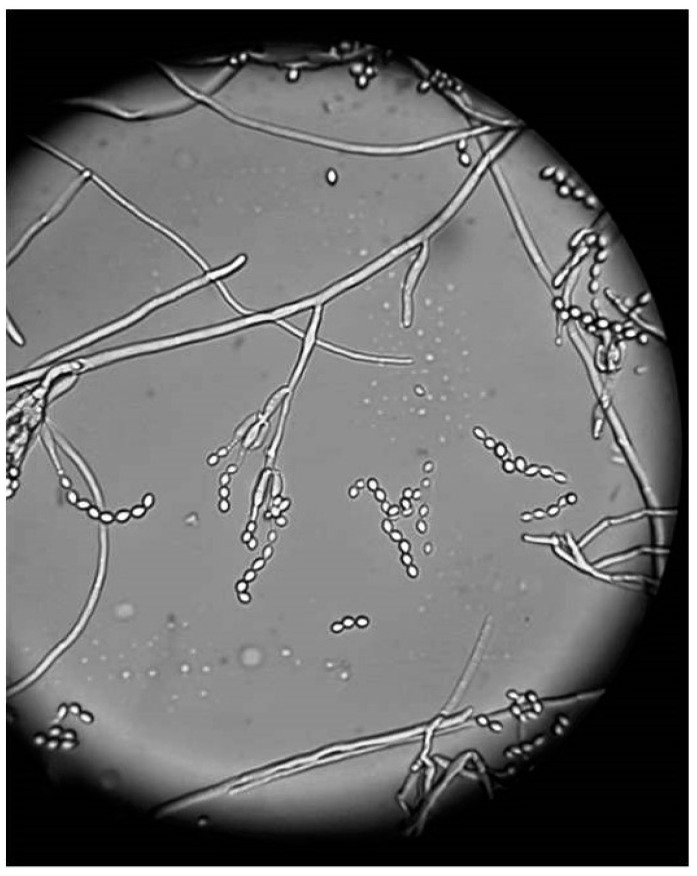
Microscopic appearance of *P. lilacinum* isolated from culture on Sabouraud agar. Magnification × 1000.

**Figure 11 pathogens-10-01046-f011:**
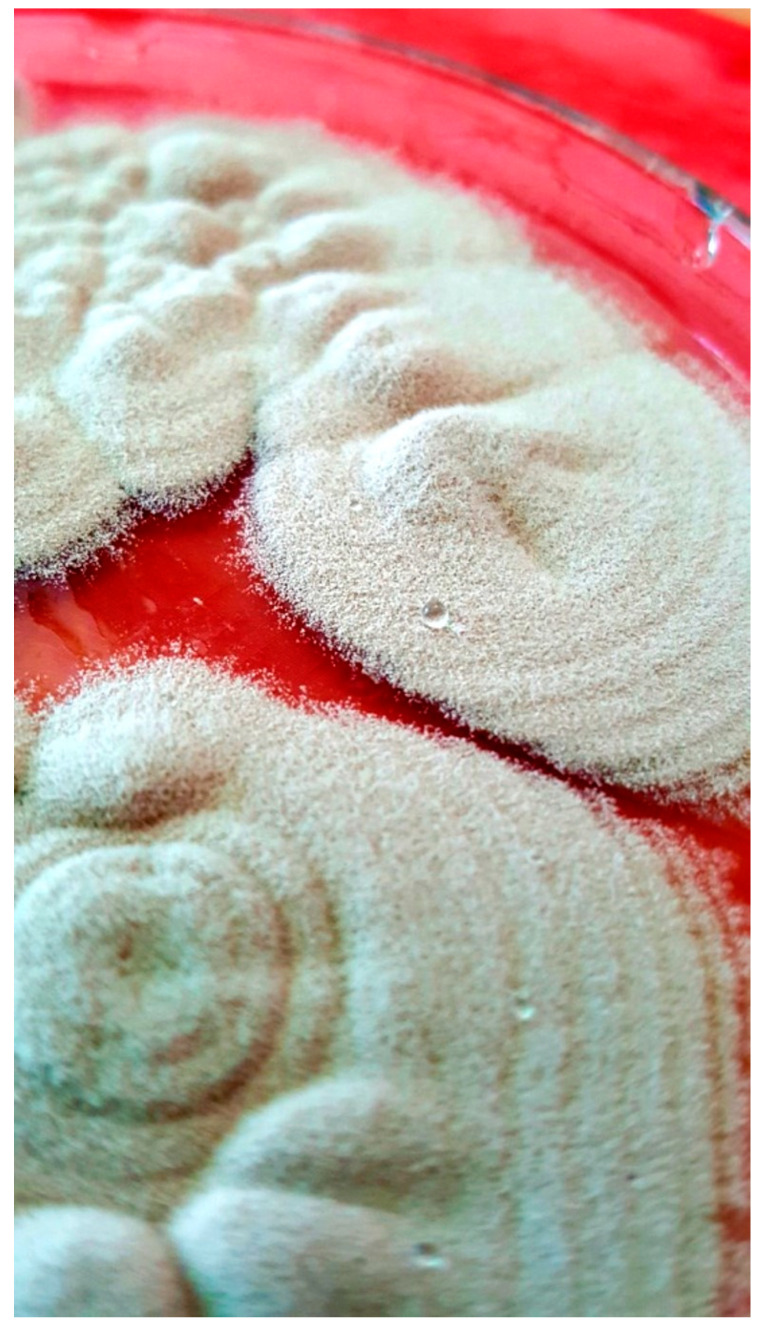
Growth of *P. lilacinum* on Sabouraud agar.

**Figure 12 pathogens-10-01046-f012:**
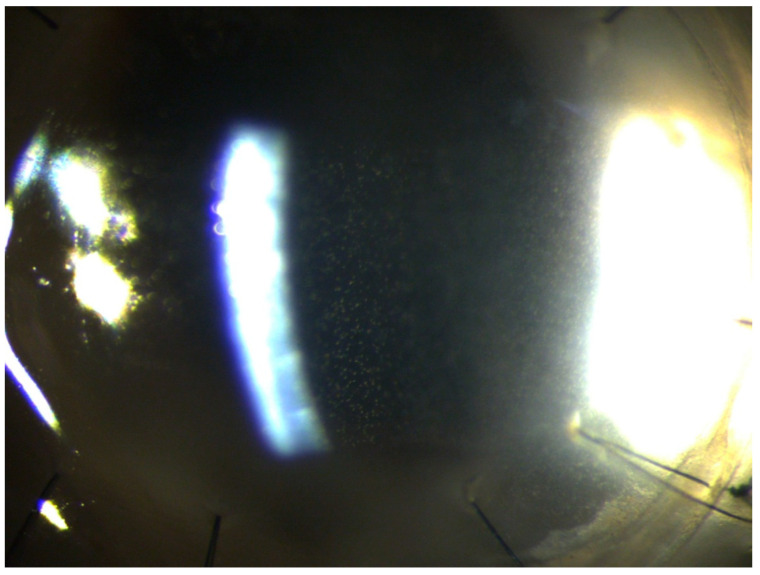
Inflammatory cells and protein exudate in the anterior chamber.

**Figure 13 pathogens-10-01046-f013:**
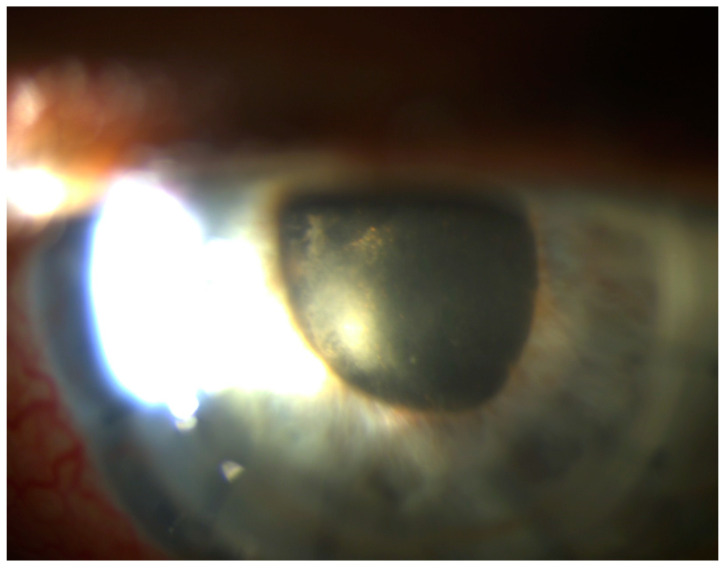
The inflammatory fibrinous membrane on the artificial lens.

## Data Availability

Not applicable.

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
