# Peer review of "First Report of a Case of Ocular Infection Caused by Purpureocillium lilacinum in Poland"

_pathogens, 2021, doi:10.3390/pathogens10081046_

Round 1
Reviewer 1 Report
In the current study, the authors presented the case report of Purpureocillium lilaci induced ocular infection. The patient developed subacute keratitis and progressive granulomatous uveitis. The authors used the culture and MALDI-TOF MS, for the identification of pathogens. A drug susceptibility assay was also carried to check the sensitivity to available drugs.
Here, are the major comments for the current report.
1) The MALDI-TOF MS used for the identification of the pathogen is after the culture of the pathogen? Did the authors use the MALDI-TOF MS directly from the patient sample? The time required for the diagnosis is increased if it is used from the culture (culture incubation can take prolonged time). In the current diagnosis, the multiplex PCR can be directly performed from the patient's sample and could be the best option to diagnose the pathogen (bacterial, viral, or fungal pathogen).
2) Authors can add the detailed method of MALDI-TOF MS along with analysis (spectra, scores etc.) or any detail important to the readers.
3) Authors can add the scale bar on all the images.
4) Authors can also add the microscopic images of Purpureocillium lilacinum.
Author Response
Dear Madam/Sir,
I and the co-authors appreciate the time and effort that you dedicated to providing feedback on our manuscript and are grateful for the insightful comments on and valuable improvements to our paper.
With regard to point 1 of your review
1) The MALDI-TOF MS used for the identification of the pathogen is after the culture of the pathogen? Did the authors use the MALDI-TOF MS directly from the patient sample? The time required for the diagnosis is increased if it is used from the culture (culture incubation can take prolonged time). In the current diagnosis, the multiplex PCR can be directly performed from the patient's sample and could be the best option to diagnose the pathogen (bacterial, viral, or fungal pathogen). the MALDI-TOF MS based pathogen identifiaction was performed from the culture.
I agree that application of PCR would provide faster identification, but this procedure had not been avaliable for us at the time of diagnosis, and also according to the local procedures all specimens must be cultured, either on solid or liquid media. Additionally the amount of specimen colleced was minute, so due to limited diagnostic procedures we decided to culture it only.
With regard to point 2 of your review
2) Authors can add the detailed method of MALDI-TOF MS along with analysis (spectra, scores etc.) or any detail important to the readers.
The information about the data acqusition procedure have been added and the MS analysis spectra have been added as Figure 1.
"[...] Mass spectra within a range of 2,000 to 20,000 Da were recorded in linear positive mode at frequency of 50 Hz by a 337 nm nitrogen laser, fixed focus. For each interrogation, laser shots at different positions within the target well produce up to 100 mass profiles. [...]"
In case of Vitek MS no scores are presented during analysis, the identification is given as a confidence value. In our study the confidence value was 99. It is stated in the section 2.3. Results.
"The VITEK MS analysis confirmed the presence of P. lilacinum (confidence value=99). [...]".
With regard to point 3 of your review
3) Authors can add the scale bar on all the images.
We agree with your suggestion that the scale bar could be added but all the picture were taken without scale bar, so it is not possible to add ones.
With regard to point 4 of your review
4) Authors can also add the microscopic images of Purpureocillium lilacinum.
We think this is an excellent suggestion. As you requested new photos has been added to the manuscript.
"The microscopic appearance of P. lilacium is presented in Figures 8-10. The pictures of culture of P. lilcinum on the Sabouraud agar is presented in Figure 11."
Reviewer 2 Report
The manuscript describes a case of Purpureocilium lilacinum keratitis and subsequent sequellae.
Introduction - first paragraph. The middle section of this - about current routine microbiology methods requires a few references
The 3rd paragraph on ocular fungal infection requires references
There are plenty of references for this statement "Human exposure to fungus-contaminated plants or other organic matter may facilitate exogenous infections of the eye resulting in corneal trauma" add at least one
The following paragraph that only has one reference (6) should have more inserted in to the body of the paragraph
Same requirement for the paragraph that ends with the single reference (8)
Results
I do not understand this "was susceptible to voriconazole (0.032 mg/L), itraconazole (32.0 mg/L), and amphotericin B (32.0 mg/L), and was interpreted as being susceptible, resistant, and resistant, respectively." please rephrase
Discussion "Many authors have suggested that even appropriate fungal drugs might not provide the desired therapeutic benefits in a short period of time, resulting in 257 their early exclusion or treatment changes. " requires references to back up the statement
Author Response
Dear Madam/Sir,
I and the co-authors appreciate the time and effort that you dedicated to providing feedback on our manuscript and are grateful for the insightful comments on and valuable improvements to our paper.
With regard to your coments and suggestions to the Introduction section:
Introduction - first paragraph. The middle section of this - about current routine microbiology methods requires a few references
The 3rd paragraph on ocular fungal infection requires references
There are plenty of references for this statement "Human exposure to fungus-contaminated plants or other organic matter may facilitate exogenous infections of the eye resulting in corneal trauma" add at least one
The following paragraph that only has one reference (6) should have more inserted in to the body of the paragraph
Same requirement for the paragraph that ends with the single reference (8)
In all indicated places new references have been added. The changes are indicated in the manuscript.
With regard to your coments and suggestions to the Results section:
I do not understand this "was susceptible to voriconazole (0.032 mg/L), itraconazole (32.0 mg/L), and amphotericin B (32.0 mg/L), and was interpreted as being susceptible, resistant, and resistant, respectively." please rephrase
Thank you for pinting this out, the sentence have been rephrased as follow:
"[...] The fungal pathogen responsible for infection was susceptible to voriconazole (0.032 mg/L), resistant to itraconazole (32.0 mg/L), and resistant to amphotericin B (32.0 mg/L).[...]"
With regard to your coments and suggestions to the Discussion section:
Discussion "Many authors have suggested that even appropriate fungal drugs might not provide the desired therapeutic benefits in a short period of time, resulting in 257 their early exclusion or treatment changes. " requires references to back up the statement.
As indicated new references have been added. The changes are indicated in the manuscript.